An integrative decision-making framework to guide policies on regulating ChatGPT usage

Bukar Umar Ali 1
http://orcid.org/0000-0002-0052-4870 Sayeed Md Shohel 1 shohel.sayeed@mmu.edu.my
http://orcid.org/0000-0002-6108-3183 Razak Siti Fatimah Abdul 1
Yogarayan Sumendra 1
Amodu Oluwatosin Ahmed 2
1 Centre for Intelligent Cloud Computing (CICC), Faculty of Information Science & Technology, Multimedia University , Melaka , Malaysia
2 Information and Communication Engineering Department, Elizade University , Ilara-Mokin, Ondo State , Nigeria
Omicini Andrea
Electronic publication date: 2024 Feb 29
Publication date: 2024
Volume: 10
Electronic Location ID: e1845
Received 2023 Aug 9; Accepted 2024 Jan 9
Copyright: © 2024 Bukar et al.
Copyright year: 2024
Copyright holder: Bukar et al.
License: This is an open access article distributed under the terms of the Creative Commons Attribution License, which permits unrestricted use, distribution, reproduction and adaptation in any medium and for any purpose provided that it is properly attributed. For attribution, the original author(s), title, publication source (PeerJ Computer Science) and either DOI or URL of the article must be cited.
License URL: https://creativecommons.org/licenses/by/4.0/

Keywords: Generative AI, ChatGPT, Ethics, Policy making, Decision making, Risk, Reward, Resilience, Systematic review, Higher education

Funding: Research Management Center (RMC) of Multimedia University Malaysia (MMU) MMUI/220159 This study is supported by the Research Management Center (RMC) of Multimedia University Malaysia (MMU) (Project No. MMUI/220159) which funded the publication fee for this article. The funders had no role in study design, data collection and analysis, decision to publish, or preparation of the manuscript.

==============================
Generative artificial intelligence has created a moment in history where human beings have begin to closely interact with artificial intelligence (AI) tools, putting policymakers in a position to restrict or legislate such tools. One particular example of such a tool is ChatGPT which is the first and world's most popular multipurpose generative AI tool. This study aims to put forward a policy-making framework of generative artificial intelligence based on the risk, reward, and resilience framework. A systematic search was conducted, by using carefully chosen keywords, excluding non-English content, conference articles, book chapters, and editorials. Published research were filtered based on their relevance to ChatGPT ethics, yielding a total of 41 articles. Key elements surrounding ChatGPT concerns and motivations were systematically deduced and classified under the risk, reward, and resilience categories to serve as ingredients for the proposed decision-making framework. The decision-making process and rules were developed as a primer to help policymakers navigate decision-making conundrums. Then, the framework was practically tailored towards some of the concerns surrounding ChatGPT in the context of higher education. In the case of the interconnection between risk and reward, the findings show that providing students with access to ChatGPT presents an opportunity for increased efficiency in tasks such as text summarization and workload reduction. However, this exposes them to risks such as plagiarism and cheating. Similarly, pursuing certain opportunities such as accessing vast amounts of information, can lead to rewards, but it also introduces risks like misinformation and copyright issues. Likewise, focusing on specific capabilities of ChatGPT, such as developing tools to detect plagiarism and misinformation, may enhance resilience in some areas (e.g., academic integrity). However, it may also create vulnerabilities in other domains, such as the digital divide, educational equity, and job losses. Furthermore, the finding indicates second-order effects of legislation regarding ChatGPT which have implications both positively and negatively. One potential effect is a decrease in rewards due to the limitations imposed by the legislation, which may hinder individuals from fully capitalizing on the opportunities provided by ChatGPT. Hence, the risk, reward, and resilience framework provides a comprehensive and flexible decision-making model that allows policymakers and in this use case, higher education institutions to navigate the complexities and trade-offs associated with ChatGPT, which have theoretical and practical implications for the future.

Introduction

The advancement of ChatGPT has thrust humanity into the era of the progress-ethics conundrum (Taecharungroj, 2023). Educational stakeholders hold divergent views on the technological transformation brought about by ChatGPT: while some discourage students from using the tool, others remain indifferent. The authors in Dwivedi et al. (2023) reported a split opinion on whether the use of ChatGPT should be restricted or regulated while Lim et al. (2023) emphasized the potential risks associated with ChatGPT, suggesting that it could be seen as a foe and thus be subject to restrictions or even a complete ban. A significant number of stakeholders are concerned that this transformation will disrupt educational practices (Haque et al., 2022; Michel-Villarreal et al., 2023; Lim et al., 2023). Consequently, striking a balance between risk, reward, and resilience has become a central topic of discussion among stakeholders and policymakers. When considering the decision to employ ChatGPT, societal megatrends such as digitalization, urbanization, globalization, climate change, automation and mobility, global health issues, the aging population, emerging markets, and sustainability play a crucial role (Haluza & Jungwirth, 2023; Jungwirth & Haluza, 2023; Kemendi, Michelberger & Mesjasz-Lech, 2022). Given the close management of these tools at present, it is essential for stakeholders to fully grasp their implications before they become a major concern (Kooli, 2023). Masters (2023) further highlights the importance of identifying, anticipating, and accommodating the implications of ChatGPT to ensure that stakeholders can make use of artificial intelligence (AI) without compromising essential ethical principles. In the same vein, Dwivedi et al. (2023) identify several areas requiring further research, including knowledge, transparency, and ethics; the digital transformation of organizations and societies; and teaching, learning, and scholarly research. Therefore, this study aims to respond to the call for proactive action in addressing the ethical problems and opportunities presented by the use of AI (Dwivedi et al., 2023; Masters, 2023).

Moreover, as a powerful language model, ChatGPT has demonstrated its potential in various applications, ranging from customer support and content generation to educational assistance and creative writing. However, as its capabilities expand, so do the potential risks and ethical concerns associated with its use. To ensure responsible and effective implementation of ChatGPT in diverse domains, it is essential to develop an integrative framework for policymaking that aids decision-making processes. Therefore, the adoption of an integrative framework for policy making regarding ChatGPT usage is imperative to ensure the responsible, ethical, and secure deployment of the AI technology. By addressing ethical concerns, fostering accountability, and improving user experience, such a framework will lay the foundation for the widespread and beneficial integration of ChatGPT across various domains while safeguarding the well-being of users and society at large.

Notably, the question of whether to restrict or legislate ChatGPT usage (Dwivedi et al., 2023; Lim et al., 2023) reflects a judgment on how to weigh the importance of societal values and civilization. Roberts (2023) emphasizes that answering such a question satisfactorily requires considering not only the ethical issues (risk) associated with ChatGPT but also the rewards and benefits derived from its usage. Each of these elements is significant, and their interactions can result in complex and often unpredictable synergies and trade-offs. Furthermore, they are influenced by and, in turn, influence people’s ability to adapt and adjust to this transformation (resilience). For example, Michel-Villarreal et al. (2023) highlighted key challenges, opportunities, barriers, and priorities of ChatGPT for higher education, as illustrated in Fig. 1. Therefore, as ChatGPT becomes increasingly integrated into people’s lives, the adaptability of educational institutions and individuals may increase, making it challenging to restrict or ban its use, prompting the need to develop a framework that can better structure policy discussions and decision-making processes. While the existing literature provides preliminary information about the ethical issues, usage, and opportunities associated with ChatGPT (Dwivedi et al., 2023; Liebrenz et al., 2023; Tlili et al., 2023; Lee, 2023a; Pavlik, 2023; Lund et al., 2023; Zhuo et al., 2023; Lund & Wang, 2023; Ali & Djalilian, 2023; Sallam, 2023; Mhlanga, 2023), these insights and findings are intended to assist practitioners in effectively make policy for or against utilizing ChatGPT. However, the current literature lacks guidance on how practitioners can utilize this vast amount of information to inform their policy and decision-making processes efficiently and effectively. As a result, this study aims to address this critical research gap by adopting and conceptualizing a theoretical framework that can incorporate such findings and demonstrate the decision-making process for the utilization of ChatGPT.

Figure 1 Considerations of ChatGPT in higher education.

Research novelty and contributions

In this study, we explore the application of the risk, reward, and resilience (RRR) framework within the specific context of ChatGPT and its implications in the higher education domain. By dissecting the intricate dynamics of risk, reward, and resilience and the underlying drivers, our research has illuminated the complexities inherent in decision-making concerning the integration of ChatGPT within educational settings. The RRR framework thus functions as a versatile and comprehensive mental model, equipping policymakers and higher education institutions to navigate the multifaceted challenges and opportunities linked to ChatGPT. It highlights the necessity of approaching risk, reward, and resilience holistically, recognizing their interdependence and potential cascading effects. Throughout our study, we have examined a spectrum of policy responses and interventions. These include legislative measures, restrictions, capacity-building initiatives, and strategies to address issues of access and equity. These diverse interventions show the multidimensional nature of ChatGPT-related challenges, requiring a careful evaluation of second-order effect and the need for balanced strategies tailored to the specific context and objectives of higher education institutions. In essence, our study furnishes a framework, roadmap, and decision-making rules for policymakers and higher education institutions to navigate the intricate terrain of ChatGPT ethics. Similarly, the unique contribution of this research lies in its adaptability, which is readily transferable, enabling any higher education institution to shape and implement academic integrity policies responsibly. Leveraging the RRR framework, examining risk, reward, and resilience, and conducting a thorough assessment of potential policy impacts, stakeholders can collectively work toward establishing a robust, inclusive, and resilient higher education system in the era of ChatGPT.

Study aim and organization

The aim of this study is to contribute toward providing guidance for discussion and decision-making concerning the utilization of ChatGPT in the education sectors, and by extension the society. Specifically, the study has three key objectives, (1) propose a decision-making process and guiding rules for RRR application, (2) identify key elements for RRR from existing studies through systematic literature review, and (3) illustrate the practical application of RRR on ChatGPT ethics conundrum. Hence, the study is structured as follows; “Study Background and Related Work” discusses the related works as well as covers a brief overview of the integrative policy framework underpinning this study; “Methodology” explains the review methodology through the systematic approach applied in the study; “RRR Application” discusses the process and guidelines of RRR application, the extracted data from the literature for risk, reward, and resilience in accordance to the integrative policy framework, the practical application of RRR in the context of ChatGPT ethics conundrum. “Discussion and Implications” discusses the research implications, and finally “Conclusion” covers the conclusion.

Study background and related work

In the wake of the introduction of ChatGPT and the emergence of concerns surrounding its ethical implications, numerous studies have been dedicated to investigating this technology. In the sections that follow, we delve into the existing body of research, exploring related work and the driving motivations behind this study. Furthermore, the study outlines and discusses the theoretical framework from which this study is grounded.

Review of the literature

The diverse range of studies presented in Table 1 reflects the growing interest from researchers regarding AI language models, specifically ChatGPT, across multiple domains (Dwivedi et al., 2023), such as healthcare (Jungwirth & Haluza, 2023; Masters, 2023; Rao, 2023), tourism (Carvalho & Ivanov, 2023; Ivanov & Soliman, 2023), journalism and media content (Pavlik, 2023), political orientation (Rozado, 2023), chemistry community, physics and science education (Emenike & Emenike, 2023; Cooper, 2023; Yeadon et al., 2023), architecture (Kwon, 2023), and entrepreneurship (Short & Short, 2023). Firstly, one of the themes found in the existing studies is the assessment of ChatGPT’s accuracy and performance (Geerling et al., 2023; Gilson et al., 2023; Ariyaratne et al., 2023; Dwivedi et al., 2023), from generating articles to answering questions, emphasizing the need for reliable and interpretable Generative artificial intelligence (Gen-AI) responses. Ethical and legal challenges related to AI-generated content are also a recurrent topic (Lee, 2023b; Salvagno, Taccone & Gerli, 2023; Ray, 2023; Karaali, 2023; Dwivedi et al., 2023), especially in the context of academic research and education, where concerns about authenticity and academic integrity arise. Moreover, many studies explore ChatGPT’s potential benefits and capabilities (Victor et al., 2023; Farrokhnia et al., 2023; Halaweh, 2023; Kooli, 2023; Cox & Tzoc, 2023; Carvalho & Ivanov, 2023; Jungwirth & Haluza, 2023; Cascella et al., 2023; Kolides et al., 2023; Dwivedi et al., 2023), and potential to address societal megatrends (Haluza & Jungwirth, 2023), with a focus on its role in writing, research, and pedagogy, suggesting a shift in educational paradigms.

Table 1 Focus and contributions by existing studies.

Ref.	Focus or contributions of the study	Method	
Ariyaratne et al. (2023)	Assessed the accuracy of ChatGPT-generated articles.	Exploratory	
Lee (2023b)	The study focuses on the controversy surrounding the role of AI chatbots, specifically ChatGPT, as potential co-authors in academic research, emphasizing the legal and ethical challenges associated with AI-generated text.	Review	
Salvagno, Taccone & Gerli (2023)	The study investigates the potential and limitations of using AI chatbots, particularly ChatGPT, in scientific writing, emphasizing their role in assisting researchers, acknowledging ethical concerns, and the need for future regulations in this context.	Position or opinion essay	
Yan (2023)	The research investigates how ChatGPT’s text generation feature was used in a one-week L2 writing practicum and explores students’ behaviors and reflections.	Exploratory: qualitative approach	
Ray (2023)	This study provides a comprehensive review of ChatGPT, an AI chatbot technology, and its impact on various aspects of scientific research and applications across different industries.	Review	
Taecharungroj (2023)	The study employs latent Dirichlet allocation (LDA) topic modeling to identify what ChatGPT can do from Twitter text.	Text analysis	
Farrokhnia et al. (2023)	The study assessed the impact of ChatGPT, an AI tool, on education using the SWOT (Strengths, Weaknesses, Opportunities, Threats) analysis framework.	SWOT analysis	
Grünebaum et al. (2023)	The study evaluated ChatGPT, with a specific emphasis on its ability to handle clinical-related queries in the field of obstetrics and gynecology.	Exploratory	
Su & Yang (2023)	The study introduces a theoretical framework called “IDEE” (Identify, Determine, Ensure, Evaluate) for the implementation of educative AI. The framework involves identifying desired educational outcomes, determining the appropriate level of automation, ensuring ethical considerations, and evaluating the effectiveness of using ChatGPT and other generative AI in education.	Conceptual	
Halaweh (2023)	The article addressed concerns and explored the potential use of ChatGPT in educational settings. It seeks to (i) make a case for integrating ChatGPT into education and (ii) offer educators a set of strategies and techniques for implementing ChatGPT responsibly and effectively in teaching or research.	Position or opinion essay	
Kooli (2023)	The article explored the potential use of AI systems and chatbots in the academic field and their impact on research and education, with a particular emphasis on the ethical perspective.	Qualitative exploratory research	
Cox & Tzoc (2023)	The article introduced ChatGPT and provided an overview of its capabilities.	Position or opinion essay	
Karaali (2023)	The study focuses on ChatGPT and the concerns it raises among educators.	Position or opinion essay	
Rao (2023)	This study highlights the potential benefits of integrating AI in healthcare.	Position or opinion essay	
Cotton, Cotton & Shipway (2023)	The article explores the opportunities and challenges associated with the use of ChatGPT in this educational setting.	Position or opinion essay	
Carvalho & Ivanov (2023)	The article examined the applications, benefits, and risks of ChatGPT and large language models in the context of the tourism sector.	Review	
Jungwirth & Haluza (2023)	The study investigates the potential for AI to automate data analysis, generate new insights, and assist in the discovery of new knowledge in the field of public health. It outlines the top 10 contribution areas of AI in public health.	Exploratory	
Pavlik (2023)	The article discusses the potential transformation of journalism and media content through Gen-AI, which outlines and discusses the impact of ChatGPT.	Position or opinion essay	
Geerling et al. (2023)	The study evaluated ChatGPT’s performance on the Test of Understanding in College Economics (TUCE), which is a standardized test of economics knowledge in the United States, primarily targeting principles-level understanding.	Exploratory	
Masters (2023)	The study focuses on AI technology which centers on the ethical concerns that Health Professions Education (HPE) teachers and administrators may encounter as they incorporate AI systems into their teaching environment.	Guidelines	
Gilson et al. (2023)	The study evaluated the performance of ChatGPT on questions within the scope of the United States Medical Licensing Examination (USMLE) Step 1 and Step 2 exams, which analyses the interpretability of ChatGPT’s responses.	Exploratory, experiment	
Cascella et al. (2023)	This article explores the potential applications and limitations of ChatGPT as well as the feasibility of using ChatGPT in clinical and research settings, considering several areas:	Exploratory	
Ivanov & Soliman (2023)	This article explores the potential impact of ChatGPT on tourism education and research, which offers insights into the transformative role of ChatGPT in these domains.	Conceptual analysis, position	
Rozado (2023)	This research investigated the political orientation of ChatGPT by subjecting it to 15 different political orientation tests.	Exploratory, experiment	
Emenike & Emenike (2023)	The article discusses the potential impacts of ChatGPT in the context of the chemistry community.	Commentary	
Thurzo et al. (2023)	The study examined the impact of AI on dentistry, particularly in the context of dental education.	Review	
Kwon (2023)	This article discusses the potential applications and implications of using AI, particularly ChatGPT, in the field of architecture.	Technical notes	
Cooper (2023)	The study explored the potential of ChatGPT in the field of science education.	Exploratory	
Kolides et al. (2023)	The article examined and analyzed the transformative potential, capabilities, and societal implications of foundation models (FMs) in AI, emphasizing the need for multidisciplinary collaboration and responsible use in comprehending and addressing their societal impact.	Review/survey	
Short & Short (2023)	The article investigated the role of ChatGPT, in shaping entrepreneurial rhetoric. The study employs a framework that categorizes CEO rhetoric into Creator, Transformer, Rebel, and Savior.	Exploratory, experiment	
Schöbel et al. (2023)	This article presents a bibliometric study that examines the evolution of research on conversational agents (CAs) over time. The authors analysed over 5,000 research articles on CAs to understand the development of technical capabilities and research paradigms.	Bibliometric analysis	
Dwivedi et al. (2023)	The article compiles insights from 43 contributions by experts from diverse fields, including computer science, marketing, information systems, education, policy, hospitality and tourism, management, publishing, and nursing.	Exploratory	
Victor et al. (2023)	The article explores the potential of ChatGPT, to address core issues linked to the Association of Social Work Boards (ASWB) licensing exams, particularly the use of a multiple-choice format that is considered to be disconnected from real-world social work practice.	Exploratory, experiment	
Yeadon et al. (2023)	The primary focus of this work is to demonstrate the capability of advanced AI language models like ChatGPT and DaVinci-003 to generate high-quality, original short-form Physics essays of 300 words within seconds.	Exploratory	
Haluza & Jungwirth (2023)	This article explores the potential of ChatGPT, to address societal megatrends such as digitalization, urbanization, globalization, climate change, automation, mobility, global health issues, and the aging population, as well as emerging markets and sustainability.	Exploratory	
Lim et al. (2023)	The study provides a comprehensive study by defining Gen-AI and transformative education, establishing the paradoxes inherent to Gen-AI, and offering implications for the future of education, particularly from the perspective of management educators.	Critical analysis through paradox theory	
Tlili et al. (2023)	This study focuses on the examination of ChatGPT in the context of education among early adopters, which explores the public discourse on social media regarding ChatGPT’s use in education as well as investigates user experiences through ten different educational scenarios.	Qualitative instrumental case study, text analysis	

Furthermore, the majority of the studies encompass exploratory approaches (Haluza & Jungwirth, 2023; Yeadon et al., 2023; Victor et al., 2023; Dwivedi et al., 2023; Rozado, 2023; Cascella et al., 2023; Ariyaratne et al., 2023; Yan, 2023; Grünebaum et al., 2023; Kooli, 2023; Jungwirth & Haluza, 2023; Geerling et al., 2023; Gilson et al., 2023; Cooper, 2023; Short & Short, 2023). Similarly, several studies take the form of reviews, such as Ray (2023), Lee (2023b), Thurzo et al. (2023), Carvalho & Ivanov (2023) and Kolides et al. (2023). Moreover, text analysis is employed in studies conducted by Taecharungroj (2023) and Tlili et al. (2023). Also, Farrokhnia et al. (2023) conducts a SWOT analysis, while several other works are opinion, position, technical notes, and commentary essays (Salvagno, Taccone & Gerli, 2023; Lee, 2023b; Halaweh, 2023; Cox & Tzoc, 2023; Karaali, 2023; Rao, 2023; Cotton, Cotton & Shipway, 2023; Pavlik, 2023; Emenike & Emenike, 2023). Finally, Kwon (2023) presents technical notes, and Schöbel et al. (2023) delves into bibliometric analysis concerning research on conversational agents.

The analysis of the literature has shown that there is a notable gap in the existing studies demonstrating the actual impact of ChatGPT on educational outcomes. As AI language models become more pervasive, the broader societal impact on information dissemination, content moderation, and public attitudes needs closer examination. Therefore, it is essential for future research to explore regulations, guidelines, and ethical frameworks to ensure the responsible integration of Gen-AI in various professional fields while addressing both advantages and challenges. Overall, the prior studies presented in Table 1 lay the foundation for in-depth investigations, emphasizing the transformative potential of Gen-AI, ethical considerations, and the need for a well-defined regulatory framework to guide future research and practical applications.

In particular, few studies have proposed theories or frameworks, but notable exceptions include the works of Su & Yang (2023) and Lim et al. (2023). The sooner proposed a theoretical framework called “IDEE” (Identify, Determine, Ensure, Evaluate) for the implementation of educative AI. The framework involves identifying desired educational outcomes, determining the appropriate level of automation, ensuring ethical considerations, and evaluating the effectiveness of using ChatGPT and other Gen-AI in education (Su & Yang, 2023). The latter provides a critical analysis through paradox theory by defining Gen-AI and transformative education, establishing the paradoxes inherent to Gen-AI (Lim et al., 2023). Nevertheless, there is a lack of existing studies that appear to have explicitly focused on creating a comprehensive decision-making tool that can assist stakeholders in all sectors in determining whether Gen-AI should be legislated or restricted. While the studies touch upon various aspects of AI, including its impact on education, ethics, and specific domains like healthcare and journalism, none of them seem to provide a unified framework or tool that guides policymakers, educators, researchers, and professionals across different fields in making informed decisions about Gen-AI usage in education. This notable gap suggests that there is a need for an interdisciplinary effort to develop a systematic and adaptable decision-making tool that can be applied across sectors to address the complex challenges and opportunities presented by Gen-AI effectively. Such a tool could help ensure the responsible and balanced integration of Gen-AI while taking into account the specific needs and concerns of various stakeholders.

Overview of RRR

Roberts (2023) introduced a new framework called risk, reward, and resilience, coined as RRR, which combines knowledge from various disciplines and fields. This framework offers a straightforward yet adaptable mental decision-making model that can be applied to diverse domains. To enhance the chances of survival and success in a complex and uncertain environment, RRR emphasizes the importance of considering the interconnections of three key factors: risk, reward, and resilience. Each of these factors consists of three drivers. Risk is determined by the combination of hazard or threat, exposure, and vulnerability. The reward is influenced by opportunity, access, and capability. Resilience, on the other hand, depends on absorptive, adaptive, and transformative capacities (see Fig. 2). According to Roberts (2023), the risk lies at the point where threat, exposure, and vulnerability intersect. This intersection relies on a combination of the severity of an external hazard or threat and how it interacts with the exposure and vulnerability of a particular entity or system. On the other hand, reward is determined by factors like opportunity, access, and capability. These elements describe the potential benefits that can be gained from a specific action, the circumstances and avenues through which an entity or system can exploit these opportunities, and the internal attributes of the entity or system that influence the gains they are likely to obtain from accessing these opportunities. In the context of RRR, risk primarily concerns negative internal and external aspects (vulnerability and threats), while rewards are associated with positive internal and external factors (capability and opportunity). As a result, capability and vulnerability are used interchangeably with strength and weakness and have broader applicability. RRR framework introduces access as a significant driver for reward, particularly relevant for policymakers whose regulations often impact access.

Figure 2 The Roberts (2023) risk, reward, and resilience framework (RRR).

Furthermore, the core of resilience is found in the ability of entities and systems to absorb, adapt to, and transform in response to ongoing changes (Béné et al., 2012; Roberts, 2023). The increased focus on resilience thinking should be seen as a response to the challenges of operating, conducting business, or governance in an increasingly complex world (Walker & Salt, 2012; Roberts, 2023). The drivers of resilience represent the dynamic capabilities that enable entities and systems to deal with change by absorbing, adapting to, and transforming in the face of threats or hazards (Roberts, 2023). Absorption refers to an entity or system’s capacity to withstand threats or hazards without experiencing significant negative consequences. Adaptation involves the ability of a system to respond to threats or hazards by making adjustments that enable it to continue functioning, albeit in a slightly altered manner. Transformation, on the other hand, pertains to the ability to alter the structures and incentives of the entity or system in a way that not only allows recovery from shocks but also fundamentally changes the entity or system for the future.

The RRR framework incorporates cross-cutting elements that can be applied across multiple domains. Within this framework, risks can be economic, security-related, or health-related. Rewards can take the form of monetary, diplomatic, or social benefits. Resilience can be physical, psychological, or environmental in nature. The RRR framework has the potential to break down silos and promote communication across disciplines, thereby enhancing policy outcomes. It offers a more comprehensive and unbiased approach compared to existing frameworks (Roberts, 2023). By identifying the drivers of each element, understanding their connections, and recognizing the policy choices and consequences associated with them, the RRR framework provides a simplified and structured systems model for tackling complex problems. It explicitly addresses the objectives, trade-offs, and assumptions that underlie policy-making processes. Importantly, the RRR framework does not dictate what individuals or policymakers should think about complex problems. Instead, it serves as a guide to help them understand how to approach such problems. It allows for the inclusion of diverse and sometimes conflicting hypotheses on a single diagram, enabling experts from different fields to see that their insights and values are respected. Simultaneously, it highlights the existence of alternative perspectives that should be evaluated and considered. In this way, the RRR framework facilitates the decision-making process by helping identify the best feasible outcome among various choices.

Methodology

The aim of this study is to propose a policy and decision-making framework, known as RRR, for the application of ChatGPT in higher education. This framework includes specific elements related to risk, reward, and resilience, which are used to demonstrate the practical applicability of the framework. The identification of data based on the RRR framework for ChatGPT is obtained from the literature through a systematic literature review (SLR) to ensure rigor and thoroughness of the search process (Cook et al., 1997; Kitchenham & Charters, 2007; Snyder, 2019; Bukar et al., 2020, 2022; Qasem et al., 2019). This is because the SLR adheres to a predetermined plan or protocol in which the criteria are explicitly defined before the review process begins. Key criteria include defining the research questions and keywords.

Accordingly, this study builds upon the research techniques and procedures employed in previous SLR studies (refer to Snyder, 2019; Bukar et al., 2020, 2022; Qasem et al., 2019; Sneesl et al., 2022b). It applies a similar strategy, adhering to established guidelines and leveraging prior experience. Numerous databases, including Google Scholar, Web of Science (WoS), and Scopus, offer access to research articles. While Google Scholar is more comprehensive in its coverage, WoS and Scopus prioritize the quality of the journals they index. The Scopus database was chosen as the primary data source for this analysis for several reasons, including its credibility and its preference for established peer-reviewed journals. Scopus encompasses most of the content found in WoS and offers a comprehensive array of references, abstracts, and summaries in line with accepted practices (Fink, 2019). In addition, Scopus hosts an impressive collection, boasting more than 87 million articles and access to over 25,000 active journal titles. This database provides the most extensive coverage of abstracts across various academic disciplines. Its reputation as a reliable resource for accessing global academic knowledge is reinforced by regular updates (Vaio, Hassan & Alavoine, 2022). Furthermore, the Scopus h-index tool, which categorizes books, authors, and journals, is a valuable feature that should not be overlooked (Hirsch, 2010), which allows this study to differentiate between types of articles in the early stage to ascertain the depth of research on ChatGPT. Since this study adopted the concept of the SLR approach, the method involves the identification, selection, and thorough evaluation of research to address the research question (Kitchenham & Charters, 2007). In this case, we aim to determine the themes of ChatGPT based on risk, reward, and resilience. Hence, the following section discusses the key steps of the SLR process for this study.

Research keyword and search process

Research keywords are determined when the systematic review protocol is established, shaping the scope of the retrieved materials (Thornley & Gibb, 2009; Okoli, 2015). These keywords were instrumental in identifying the articles (Ridley, 2012). As a result, relevant research keywords were formulated and employed in the search query to explore articles pertaining to the area of interest in the Scopus database. The selection of research keywords adhered to the guidelines outlined by Okoli (2015), emphasizing the need for researchers to transparently and comprehensively specify their chosen keywords. These keyword terms encompassed phrases like “ChatGPT,” “large language models,” “LLMs,” “ethical issues,” “concerns,” “ethics,” and “implications,” with various combinations such as ChatGPT OR large language models OR LLMs AND “ethical issues” OR concerns OR ethics OR implications; or ChatGPT AND “ethical issues” OR concerns OR ethics OR implications. On 3rd May 2023, a comprehensive search was conducted on Scopus database to gather relevant scholarly articles related to the study’s focus on ChatGPT. The search used the selected keywords, depicted in Fig. 3. The initial search yielded a total of 74 results. To narrow down the search to the most relevant articles, the keywords were specifically limited to ChatGPT, resulting in 47 articles. Subsequently, an Excel file was obtained from Scopus, containing the titles and abstracts of these 47 articles. Each article’s title and abstract were carefully reviewed to ascertain its suitability for inclusion in the study.

Figure 3 Review method through systematic approach.

Article selection

The initial phase involved the screening of titles and abstracts as the primary criteria to identify and eliminate irrelevant articles. This process assisted the researchers in determining whether the articles aligned with the predefined inclusion or exclusion criteria. The exclusion criteria encompassed non-English content, conference articles, book chapters, and editorial pieces, while the inclusion criteria comprised all articles listed in Scopus. To ensure the data’s quality and relevance, the selection of articles indexed in Scopus ensured that they met the quality assessment standards established for this study. Finally, undertaking the screening process eliminated six articles that were either editorials or not directly aligned with the study’s focus. This step was aimed at maintaining the integrity and coherence of the final dataset (Kitchenham & Charters, 2007; Ahmed et al., 2019). Ultimately, a total of 41 articles were deemed suitable and included as the final study sample. Consequently, these 41 articles were chosen for data extraction, as illustrated in Fig. 3.

Data extraction and analysis

Data extraction served as a means for researchers to obtain specific information from the articles. Thus, the selected articles were subjected to a systematic data extraction process, focusing on the three key terms of the RRR framework: risk, reward, and resilience. Accordingly, data was extracted from the articles according to the aim of the study. The information encompassed themes or topics related to these elements. The objective of data synthesis was to systematically organize and classify the various themes identified in the articles. To achieve this, a thematic analysis approach was employed (Clarke, Braun & Hayfield, 2015; Abedin, Jafarzadeh & Olszak, 2021; Babar, Bunker & Gill, 2018). The study closely scrutinized the information to identify recurring themes, including common topics, ideas, patterns, and approaches. A primary theme was established when the extracted data was found to be related to other sub-themes that shared a similar logical context. For instance, themes like copyright, compliance with copyright laws, consent, and legal issues were grouped together, as well as themes like misinformation, inaccuracy of information, and nonsensical content were grouped together. This process facilitated the researchers in comprehending the diverse themes reported in existing studies. Additionally, the information was assessed and the frequency of terminology was computed to illustrate their occurrence in the literature, which was done through critical analysis as presented in Tables A1–A3 (refer to Appendix A).

Table A1 Critical analysis of risk-related elements.

Reference	Private data input, confidentiality	Quality of output	Misinformation, nonsense content	Bias response	Cybersecurity	Academic integrity and honesty concern	Plagiarism	Job expectations and evolution	Copyright, compliance, consent, legal issues	Incorrect citations practices	Ownership, authorship, impersonation	Infodemics	Lack of originality	Transparency	Reliability, factual reliability	Self-reliant and lazy	Deception, manipulation, mislead, risk of hallucination	Digital divide, perpetuating discrimination in education	Educational equity	Safety issues or security from misuse	Lack of deep understanding	Declining high-order cognitive and thinking skills	Outdated data, limited knowledge	Exploitation	Responsibility	Autonomy	Beneficence	Anonymity	Accountability	Mimic people	Fidelity	Emotion	
Dowling & Lucey (2023)	✓	✓																															
Eggmann et al. (2023)	✓		✓	✓	✓																												
Perkins (2023)						✓	✓																										
Lund et al. (2023)							✓	✓	✓	✓	✓																						
Sallam (2023)				✓	✓		✓		✓	✓		✓	✓	✓			✓						✓										
Valentín-Bravo et al. (2023)			✓												✓																		
Qasem (2023)							✓									✓																	
Ariyaratne et al. (2023)			✓							✓																							
Lee (2023b)									✓								✓																
Salvagno, Taccone & Gerli (2023)			✓				✓											✓															
Yan (2023)						✓													✓														
Ray (2023)				✓																✓													
Taecharungroj (2023)								✓																									
Farrokhnia et al. (2023)		✓		✓		✓	✓											✓			✓	✓											
Grünebaum et al. (2023)										✓							✓				✓		✓										
Su & Yang (2023)		✓																		✓													
Halaweh (2023)																✓						✓											
Kooli (2023)																				✓				✓									
Karaali (2023)																						✓											
Rao (2023)	✓																																
Cotton, Cotton & Shipway (2023)						✓	✓											✓															
Carvalho & Ivanov (2023)	✓		✓		✓			✓				✓	✓				✓																
Jungwirth & Haluza (2023)			✓							✓																							
Geerling et al. (2023)						✓																											
Masters (2023)	✓			✓					✓		✓			✓						✓					✓	✓	✓	✓					
Cascella et al. (2023)																				✓													
Rozado (2023)				✓																✓													
Emenike & Emenike (2023)				✓		✓	✓											✓	✓										✓				
Thurzo et al. (2023)															✓																		
Cooper (2023)									✓			✓					✓																
Short & Short (2023)																	✓													✓			
Dwivedi et al. (2023)	✓		✓	✓								✓								✓													
Victor et al. (2023)											✓																						
Yeadon et al. (2023)																															✓		
Tlili et al. (2023)	✓					✓											✓														✓	✓	
Frequency	7	3	7	8	3	7	8	3	5	5	3	4	2	2	2	2	7	4	2	7	2	3	2	1	1	1	1	1	1	1	2	1	

Table A2 Critical analysis of reward-related elements.

Reference	Assist with research, thesis, assignment, essay	Idea generation	Data identification	Decision support	Text summarization	Writing fluency and efficiency	Multilingual communication, translation	Easy access	Creating original content	Question answering	Dissemination and diffusion of new information: access to information	Streamlining workflow	Cost saving	Documentation	Personalized learning	Improved literacy	Critical thinking and problem-based learning	Advanced than search engines	Proofreading and editing	Data processing	Hypothesis generation	Code writing	Prompt writing	Decrease teaching workload	Supporting expertise and judgment	Text generation	Provide feedback	Increased student engagement	Increase productivity and efficiency	Assemble or organise text	Passed exams	Teaching and mentoring	Support professional activities	Pitches	Support societal megatrends	Collaboration and freindship	Transformation	Usefulness	
Dowling & Lucey (2023)	✓	✓	✓																																				
Eggmann et al. (2023)				✓	✓	✓	✓																																
Perkins (2023)	✓							✓	✓																														
Lund et al. (2023)		✓			✓		✓			✓	✓																												
Sallam (2023)												✓	✓	✓	✓	✓	✓																						
Ariyaratne et al. (2023)	✓																																						
Lee (2023b)																		✓																					
Salvagno, Taccone & Gerli (2023)	✓					✓													✓											✓									
Yan (2023)						✓						✓																											
Ray (2023)	✓																			✓	✓																		
Taecharungroj (2023)	✓									✓							✓					✓	✓																
Farrokhnia et al. (2023)											✓	✓			✓									✓															
Grünebaum et al. (2023)											✓																												
Su & Yang (2023)															✓									✓			✓												
Halaweh (2023)		✓																								✓													
Kooli (2023)																									✓														
Cox & Tzoc (2023)	✓									✓												✓				✓													
Karaali (2023)	✓									✓																													
Rao (2023)												✓																											
Cotton, Cotton & Shipway (2023)								✓							✓												✓	✓								✓			
Carvalho & Ivanov (2023)												✓																	✓										
Jungwirth & Haluza (2023)					✓																					✓				✓									
Geerling et al. (2023)																															✓								
Gilson et al. (2023)																															✓								
Cascella et al. (2023)	✓																												✓										
Ivanov & Soliman (2023)	✓																									✓													
Emenike & Emenike (2023)	✓																							✓					✓			✓	✓						
Thurzo et al. (2023)	✓																																						
Cooper (2023)																			✓																				
Short & Short (2023)																																		✓					
Dwivedi et al. (2023)																													✓										
Victor et al. (2023)																															✓								
Yeadon et al. (2023)	✓																																						
Haluza & Jungwirth (2023)																																			✓				
Lim et al. (2023)								✓																												y			
Tlili et al. (2023)									✓						✓																						✓	✓	
Frequency	13	3	1	1	3	3	2	3	2	4	3	5	1	1	5	1	2	1	2	1	1	2	1	3	1	4	2	1	4	2	3	1	1	1	1	2	1	1	

Table A3 Critical analysis of resilience-related elements.

Reference	Expertise input	Appropriate testing framework	Establish acceptable usage in Science	Co-creation between humans and AI; improved human-AI interaction	Academic integrity policies	Improving automated writing evaluations (AWE)	Enhancing research equity and versatility	Not a replacement for human judgment	Establishment of corresponding pedagogical adjustments	Addressing the digital divide	Potential mitigation strategies	Balance between AI-assisted innovation and human expertise	Self-improving capability	Promote responsible usage; solidifying ethical values	Use AI detector tools	Audit trail of queries	Transform educational systems	Sustainability	Raising awareness; scientific discourse	Focus on higher-level skills and habits: (quantitative literacy (QL) and quantitative reasoning (QR))	Rigorous guidelines; developing policies and procedures	Significant training and upskilling	Experimental learning framework	Reintroduce proctored, in-person assessments	Proactive action	
Dowling & Lucey (2023)	✓	✓																								
Eggmann et al. (2023)			✓																							
Perkins (2023)				✓	✓	✓																				
Sallam (2023)							✓																			
Salvagno, Taccone & Gerli (2023)	✓							✓																		
Yan (2023)		✓	✓		✓				✓																	
Ray (2023)				✓						✓	✓	✓														
Farrokhnia et al. (2023)													✓													
Grünebaum et al. (2023)														✓												
Halaweh (2023)				✓	✓									✓	✓	✓										
Kooli (2023)				✓					✓					✓			✓	✓	✓							
Karaali (2023)																				✓						
Rao (2023)																					✓	✓				
Cotton, Cotton & Shipway (2023)			✓		✓						✓			✓	✓				✓		✓	✓				
Carvalho & Ivanov (2023)				✓																✓	✓	✓				
Jungwirth & Haluza (2023)			✓		✓									✓					✓							
Geerling et al. (2023)		✓															✓			✓			✓	✓		
Masters (2023)																									✓	
Cascella et al. (2023)																				✓						
Ivanov & Soliman (2023)			✓		✓				✓								✓			✓	✓			✓		
Thurzo et al. (2023)		✓												✓			✓									
Cooper (2023)												✓		✓						✓						
Dwivedi et al. (2023)				✓										✓				✓				✓				
Victor et al. (2023)																				✓			✓	✓		
Haluza & Jungwirth (2023)																		✓								
Lim et al. (2023)											✓						✓									
Tlili et al. (2023)														✓												
Frequency	2	4	5	6	6	1	1	1	3	1	3	2	1	9	2	1	5	3	3	7	4	4	2	3	1	

Rrr application

In this section, we delve into the practical implementation of RRR within the context of ChatGPT. RRR serves as an essential framework for ensuring the ethical and responsible usage of ChatGPT. Within this framework, we address key components of the RRR application, provide guidelines for its effective use, and explore its impact on ChatGPT. Specifically, we begin by outlining the structured process of applying RRR and present a set of guidelines to facilitate responsible decision-making in the development and deployment of ChatGPT. This aspect provides a systematic approach to addressing ethical challenges and potential pitfalls. Secondly, we identify and discuss the fundamental elements of ChatGPT that are informed by the principles of RRR. These elements reflect the integration of ethical considerations and responsible practices into the system’s design and operation. Thirdly, we illustrate the practical application of RRR by examining a real-world ethics conundrum within the context of ChatGPT. This case study demonstrates how RRR can guide decision-making and provide a responsible solution to complex ethical dilemmas. Finally, we summarize the key findings and insights from this section, highlighting the significance of RRR in shaping the responsible development and use of ChatGPT. Readers can gain a comprehensive understanding of how RRR can be practically applied to ensure ethical and responsible AI implementation.

RRR application process and guidelines

Policymakers can use the RRR framework to inform their decision-making processes and improve the effectiveness of policies in various domains, such as economics, security, and environmental management. Figure 4 proposes the process of utilizing RRR for decision-making, as illustrated in three (3) steps. According to the framework proposed in this study, policymakers should identify and assess potential risks associated with a specific system. They should be able to analyze threats, vulnerabilities, and exposure related to the system. The risks are weighted and prioritized based on their severity and potential impact. This prioritization can help in determining the most critical issues. Secondly, the potential rewards to be gained from the system are assessed and evaluated, such as the potential benefits and opportunities associated with the system. This is done considering the capability and access factors that can lead to positive outcomes. The rewards are also weighted and prioritized based on their potential positive impact. Thirdly, the potential capability that can make stakeholders withstand the risks associated with the system is identified. In general policy making should be driven by maximizing the benefits and minimizing the risks and using an objective assessment of resilience in weighting the benefits and risks. As a result, our study puts forth a set of decision-making rules, outlined in Table 2, designed to guide and inform the application of the RRR framework in the creation of policies that are both effective and efficient. It is important to acknowledge that these proposed decision-making rules, inspired by Kamali (2015), can be adapted and customized to suit the specific characteristics and demands of the system under assessment for policy development.

Figure 4 Process of RRR application.

Table 2 Proposed RRR decision-making (DM) rules for policy creation.

Rule number	Focus RRR component	Scenario	
DM Rule 1	Risk and rewards	Choosing either to prevent risks or acquire rewards when preponderance cannot be given to one over another. In this case, when in doubt about decision-making as to either gaining rewards or preventing risks because both hold the same weight or risk outweighs the reward, preference may be given to avoiding the risks to avert their consequences.	
DM Rule 2	Risk	Choosing either to prevent risks of a wider scale (general) as compared to a more specific risk when a choice has to be made between the two. In this case, preventing a more general risk may be preferred over a more confined, due to the broader consequences of the general risk. E.g, one or two specialists may benefit from Generative AI on very technical subjects or issues, even in the presence of Gen-AI hallucinations (because they can identify valid AI-generated outputs), however, because those few benefit from it may not rule out its general regulation or prevention if more less-experienced people would be affected by the risk because of their lack of knowledge or ability to recognize such hallucinations.	
DM Rule 3	Risk and resilience	Choosing whether to prevent or tolerate risks. In this case, a greater risk may be prevented by tolerating a lesser one. For instance, the way to go might be to “allow” ChatGPT to act as an “assistant” in some type of academic work (e.g., coding) in certain circumstances with acknowledgment while taking full responsibility. This is better than making a policy to allow it blatantly without end users being able to recognize what was written/coded by humans or generated by AI.	
DM Rule 4	Risk and resilience	Choosing between two risks because one cannot be avoided. In this case, the smaller of the risks may be chosen to prevent the occurrence of the risk with more dire consequences.	
DM Rule 5	Risk and resilience	Choosing between curtailing and avoiding risks. In such cases, risks can be avoided as much as possible. If risks cannot be avoided, a thorough resilience mechanism may be needed to absorb the risk as much as possible.	
DM Rule 6	Risk and resilience	Choosing whether to remedy the greater risk through adaptation or confinement. In this case, if the greater risk cannot be warded off or avoided (because it is impossible to take the option of a lesser risk), then efforts must be made to reduce the risk or confine it as much as possible.	
DM Rule 7	Rewards	Choosing one of two rewards. In such cases preference may be given to the decision that yields the maximum rewards. In the case that both can be accommodated, both rewards should be accrued as much as possible instead of choosing only one.	

Accordingly, the decision-making rules ensure that policies are designed with resilience in mind, focusing on the ability to absorb, adapt, and transform in response to changing conditions. This includes incorporating strategies to mitigate the impact of potential threats, such as promoting adaptability and supporting transformation. Moreover, the guiding rules encourage systems and organizations to adapt and adjust in response to unexpected challenges, making them more robust and capable of withstanding disruptions. In addition, the decision-making rules are designed to facilitate the transformation of people, institutions, or organizations to better address emerging challenges or opportunities. Notably, the RRR framework encourages a dynamic approach to policy-making. The decision-making rules are created to continually monitor the evolving risk landscape, adapt policies as needed, and remain flexible in response to changing circumstances. By applying the RRR framework, policymakers can create more robust and effective policies that not only consider potential risks and rewards but also build resilience in the people or society. This approach can lead to more adaptive and successful policy outcomes in an ever-changing and complex world.

Elements of ChatGPT based on RRR

As a way to help decision-makers make an effective decision regarding LLM tools, this study makes an effort to identify risk, reward, and resiliency aspect associated with LLMs by using the RRR framework (Roberts, 2023), which was identified from various elements of ChatGPT in literature. Accordingly, this information was extracted based on the three themes of the integrative framework. ChatGPT usage was evaluated through the RRR framework to provide a comprehensive report to guide stakeholders and policymakers on the future utilization of ChatGPT, by extension LLMs tools. The proceeding sections discuss these themes and their critical analysis from the literature.

Risk

The list of terms related to “Risks” highlights various challenges and potential negative implications associated with the adoption of generative AI systems. These risks encompass concerns such as privacy, data confidentiality (Dowling & Lucey, 2023; Eggmann et al., 2023; Dwivedi et al., 2023; Rao, 2023; Carvalho & Ivanov, 2023; Masters, 2023; Tlili et al., 2023), output quality (Dowling & Lucey, 2023; Farrokhnia et al., 2023; Su & Yang, 2023), bias (Eggmann et al., 2023; Sallam, 2023; Ray, 2023; Farrokhnia et al., 2023; Masters, 2023; Rozado, 2023; Emenike & Emenike, 2023; Dwivedi et al., 2023), misinformation (Eggmann et al., 2023; Valentín-Bravo et al., 2023; Ariyaratne et al., 2023; Salvagno, Taccone & Gerli, 2023; Carvalho & Ivanov, 2023; Jungwirth & Haluza, 2023; Dwivedi et al., 2023), cybersecurity (Eggmann et al., 2023; Sallam, 2023; Carvalho & Ivanov, 2023), academic integrity (Perkins, 2023; Yan, 2023; Farrokhnia et al., 2023; Cotton, Cotton & Shipway, 2023; Geerling et al., 2023; Emenike & Emenike, 2023; Lim et al., 2023), job evolution (Lund et al., 2023; Taecharungroj, 2023; Carvalho & Ivanov, 2023), copyright compliance (Lund et al., 2023; Sallam, 2023; Lee, 2023b; Masters, 2023; Cooper, 2023), ownership (Lund et al., 2023; Masters, 2023; Victor et al., 2023), transparency (Sallam, 2023; Masters, 2023), reliability (Valentín-Bravo et al., 2023; Thurzo et al., 2023), and ethical considerations. Notably, the deployment of AI systems necessitates paying careful attention to these risks in order to mitigate their potential adverse effects. Privacy concerns emphasize the importance of protecting user data and ensuring compliance with privacy regulations. Data confidentiality is vital to prevent unauthorized access or leakage of sensitive information.

In addition, the quality of AI system output is a significant concern, as inaccuracies, misinformation, and nonsense content can lead to unreliable or misleading results. Bias response poses the risk of perpetuating existing biases, potentially resulting in unfair or discriminatory outcomes. Cybersecurity issues emphasize the need to protect AI systems from malicious attacks or unauthorized access. Academic integrity and honesty concerns encompass risks such as plagiarism (Perkins, 2023; Lund et al., 2023; Sallam, 2023; Qasem, 2023; Salvagno, Taccone & Gerli, 2023; Farrokhnia et al., 2023; Cotton, Cotton & Shipway, 2023; Emenike & Emenike, 2023), incorrect citation practices (Lund et al., 2023; Sallam, 2023; Ariyaratne et al., 2023; Grünebaum et al., 2023; Jungwirth & Haluza, 2023), and lack of originality (Sallam, 2023; Carvalho & Ivanov, 2023). Job evolution and expectations highlight the potential impact of AI on employment and the need to address evolving job roles and expectations. Compliance with copyright laws, consent, and legal issues is crucial to ensure the ethical and legal usage of AI technologies.

Furthermore, the risks associated with ownership, authorship, impersonation, infodemics (Sallam, 2023; Carvalho & Ivanov, 2023; Cooper, 2023; Dwivedi et al., 2023), and lack of deep understanding (Farrokhnia et al., 2023; Grünebaum et al., 2023) emphasize the importance of responsible and accountable AI usage. Declining high-order cognitive and thinking skills (Farrokhnia et al., 2023; Halaweh, 2023; Karaali, 2023) raise concerns about over-reliance (Halaweh, 2023; Qasem, 2023) on AI systems, potentially leading to a decrease in critical thinking and problem-solving abilities. Accordingly, data not being apparently updated (Sallam, 2023; Grünebaum et al., 2023) can result in outdated or irrelevant information being presented by AI systems. Exploitation risks (Kooli, 2023) emphasize the need to prevent the misuse or abuse of AI technologies. Educational equity (Yan, 2023; Emenike & Emenike, 2023) concerns highlight the potential for AI to widen the digital divide if not accessible to all individuals (Salvagno, Taccone & Gerli, 2023; Farrokhnia et al., 2023; Cotton, Cotton & Shipway, 2023; Emenike & Emenike, 2023).

Additionally, safety or security issues from misuse (Yan, 2023; Su & Yang, 2023; Kooli, 2023; Masters, 2023; Cascella et al., 2023; Rozado, 2023; Dwivedi et al., 2023) underscore the importance of implementing measures to prevent AI systems from being used for malicious purposes. Lack of deep understanding emphasizes the limitations of AI systems in comprehending complex or nuanced contexts accurately. Despite the fact that AI systems offer significant benefits, it is essential to acknowledge and address the associated risks. Implementing safety measures, ethical practices, and regulatory frameworks is crucial to ensure responsible AI deployment that respects privacy, fairness, transparency, and accountability while mitigating potential negative consequences. Figure 5 demonstrates the risks and ethics-associated issues related to ChatGPT.

Figure 5 Aspect of risk associated with ChatGPT.

Reward

The list of terms related to “Reward” encompasses a wide range of benefits and positive outcomes associated with various aspects of utilizing AI, particularly in the context of text-related tasks. The key emphasis of these terms is that AI-powered systems provide numerous advantages, such as improved efficiency, effectiveness, accessibility, and support for diverse activities across different domains. Firstly, one prominent theme is the generation of ideas, which includes idea generation (Dowling & Lucey, 2023; Lund et al., 2023; Halaweh, 2023), hypothesis generation (Ray, 2023), and prompt writing (Taecharungroj, 2023). AI systems can assist in sparking creative thinking and generating innovative ideas, contributing to problem-solving and critical thinking skills (Sallam, 2023; Taecharungroj, 2023). Similarly, efficiency and productivity enhancements are highlighted through terms such as data identification (Dowling & Lucey, 2023), streamlining workflows (Sallam, 2023; Yan, 2023; Farrokhnia et al., 2023; Rao, 2023; Carvalho & Ivanov, 2023), cost savings (Sallam, 2023), and increased productivity (Carvalho & Ivanov, 2023; Cascella et al., 2023; Emenike & Emenike, 2023). AI technologies can automate time-consuming tasks, streamline processes, and save resources, enabling individuals and organizations to achieve more performance in less time.

Moreover, enhanced communication and access to information are emphasized through multilingual communication and translation services (Eggmann et al., 2023; Lund et al., 2023), and easy access to information (Perkins, 2023; Cotton, Cotton & Shipway, 2023; Lim et al., 2023). AI systems facilitate effective communication and break down language barriers, enabling users to access and understand information in their preferred language. Also, AI’s support for learning and education is evident in terms like personalized learning (Sallam, 2023; Farrokhnia et al., 2023; Su & Yang, 2023; Cotton, Cotton & Shipway, 2023; Tlili et al., 2023), improved literacy (Sallam, 2023), critical thinking and problem-based solving learning (Sallam, 2023; Taecharungroj, 2023), decreased teaching workload (Farrokhnia et al., 2023; Su & Yang, 2023; Emenike & Emenike, 2023), and teaching and mentoring activities (Emenike & Emenike, 2023). AI-powered tools can adapt to individual learning needs, enhance literacy skills, and assist in creating engaging and interactive learning environments. Hence, text-related tasks, such as text summarization (Eggmann et al., 2023; Jungwirth & Haluza, 2023; Lund et al., 2023), writing fluency and efficiency (Eggmann et al., 2023; Salvagno, Taccone & Gerli, 2023; Yan, 2023), proofreading and editing (Sallam, 2023; Cooper, 2023), text generation (Halaweh, 2023; Cox & Tzoc, 2023; Jungwirth & Haluza, 2023), and assembling or organizing text (Salvagno, Taccone & Gerli, 2023; Jungwirth & Haluza, 2023), benefit individuals and organizations from using AI technologies. These tools can automate and improve the quality of text-related tasks, saving time and effort while maintaining accuracy and coherence.

Additionally, AI’s role in decision support and expertise is highlighted through terms like decision support (Eggmann et al., 2023), supporting expertise and judgment (Kooli, 2023), providing feedback (Su & Yang, 2023; Cotton, Cotton & Shipway, 2023), and supporting professional activities (Emenike & Emenike, 2023). AI systems can provide valuable insights, assist in decision-making processes, and offer expert-level guidance and feedback. Lastly, the extracted keywords also recognize the positive impact of AI on collaboration, transformation, and societal advancements. AI-powered systems foster collaboration and friendship (Cotton, Cotton & Shipway, 2023; Lim et al., 2023), support pitches and presentations (Short & Short, 2023), and align with societal megatrends (Haluza & Jungwirth, 2023), leading to transformation and positive change. Nevertheless, the major highlight of these terms is that AI (as presented in Fig. 6), particularly in the realm of text-related tasks, offers a multitude of rewards and benefits. From idea generation to increased productivity, improved learning experiences, and enhanced decision-making, AI-powered systems contribute to efficiency, effectiveness, and positive outcomes across various domains and activities.

Figure 6 Aspect of reward associated with ChatGPT.

Resilience

The articles reviewed provided various terms associated with resilience, indicating that humanity is showcasing readiness to cope with the challenges of ChatGPT. A typical representation of the resilience elements can be seen in Fig. 7. According to the literature, resilience, as applied to ChatGPT, encompasses a range of considerations that contribute to the model’s robustness, adaptability, ethical usage, and potential for improvement. Similarly, this study explored various terms related to resilience and their implications for the development and deployment of ChatGPT. The extracted terms related to resilience in the context of ChatGPT highlight several keywords. First, it emphasizes the importance of establishing an appropriate testing framework (Dowling & Lucey, 2023; Yan, 2023; Geerling et al., 2023; Thurzo et al., 2023) and acceptable usage guidelines in scientific research (Eggmann et al., 2023; Yan, 2023; Cotton, Cotton & Shipway, 2023; Jungwirth & Haluza, 2023; Ivanov & Soliman, 2023). This ensures that ChatGPT is rigorously evaluated and employed ethically to maintain academic integrity. The concept of co-creation between humans and AI systems emerges as a crucial aspect (Perkins, 2023; Ray, 2023; Halaweh, 2023; Kooli, 2023; Carvalho & Ivanov, 2023; Dwivedi et al., 2023), emphasizing collaboration and continuous feedback loops to refine ChatGPT’s responses and address limitations and biases. Improving human-AI interaction through natural language interfaces and ability features promotes trust and effective collaboration.

Figure 7 Resilience themes associated with ChatGPT.

Moreover, ethical considerations are central to resilience, and responsible usage is encouraged. This includes solidifying ethical values (Grünebaum et al., 2023; Halaweh, 2023; Kooli, 2023; Cotton, Cotton & Shipway, 2023; Jungwirth & Haluza, 2023; Thurzo et al., 2023; Cooper, 2023; Dwivedi et al., 2023), utilizing AI detector tools (Halaweh, 2023; Cotton, Cotton & Shipway, 2023), and establishing policies and procedures (Rao, 2023; Cotton, Cotton & Shipway, 2023; Carvalho & Ivanov, 2023; Ivanov & Soliman, 2023) to safeguard against potential risks and biases. Raising awareness about the capabilities and limitations of ChatGPT contributes to responsible usage (Kooli, 2023; Cotton, Cotton & Shipway, 2023; Jungwirth & Haluza, 2023). Resilience in ChatGPT extends beyond individual interactions and incorporates considerations of equity and versatility (Sallam, 2023). Efforts to enhance research equity, bridge the digital divide (Ray, 2023), and address diverse language and cultural variations are essential. The aim is to ensure that ChatGPT is accessible and beneficial to a wide range of users. Furthermore, the discussion emphasizes that ChatGPT should not be seen as a replacement for human judgment (Salvagno, Taccone & Gerli, 2023), but rather as a tool to support decision-making processes. Maintaining a balance between AI-assisted innovation and human expertise is vital (Ray, 2023; Cooper, 2023). Furthermore, the study highlights the potential for self-improvement within ChatGPT (Farrokhnia et al., 2023), as well as the need for continuous training, upskilling, and research to enhance its capabilities (Rao, 2023; Cotton, Cotton & Shipway, 2023; Carvalho & Ivanov, 2023; Dwivedi et al., 2023). This ensures that ChatGPT remains adaptable, versatile, and aligned with evolving needs and advancements in the field. Overall, the key highlight from these terms is that resilience in ChatGPT lies in its robustness, adaptability, ethical usage, and the collaborative partnership between humans and AI. By addressing challenges, embracing opportunities, and promoting responsible usage, ChatGPT can effectively augment various domains while aligning with societal values and needs.

Practical application of RRR on ChatGPT ethics conundrum

In the context of ChatGPT ethics, this study aims to adopt the framework (Roberts, 2023) that emphasizes the integration of risk, reward, and resilience for effective policy-making. This framework acknowledges that merely considering risk or reward in isolation is insufficient. Policymakers must internalize both elements and recognize how they are influenced by and impact resilience over time. Understanding the dynamic interplay of these elements is crucial for determining the likelihood of survival and success of actors or systems, such as AI systems like ChatGPT. While risk, reward, and resilience are interconnected, their analysis should not be conducted in isolation. Complex problems, like those related to AI ethics, involve numerous interacting variables. The RRR framework recognizes that changes in one element or driver can have positive or negative effects on others. This interconnectedness can be visualized through RRR diagrams (refer to Figs. 8–10), revealing synergies or trade-offs between risk, reward, and resilience. Additionally, effects can be unpredictable, leading to both positive and negative outcomes.

Figure 8 RRR framework showing positives effect between the risk and rewards.

Figure 9 Multifaceted effect between risk, rewards, and resilience.

Figure 10 Relationships and effect of legislative policy between risk, reward, and resilience.

When it comes to interventions and decision-making, policymakers face choices regarding risk reduction, reward enhancement, and resilience building. The RRR framework encourages a holistic approach, recognizing the equal importance of risk, reward, and resilience rather than prioritizing one over the others categorically. Depending on the context, policymakers may choose to focus on reducing risks, increasing rewards, and/or strengthening resilience, often employing a combination of these strategies. The choice of approach depends not only on the risk tolerance of the actor but also on the nature of the environment. In stable and low-risk environments, maximizing rewards may be the priority. However, in situations with increased risks and uncertainty, actors are likely to prioritize resilience-building to withstand potential shocks. The level of resilience also influences the trade-offs between risk and reward. High risks and rewards may be acceptable if resilience is high, but caution is warranted when resilience is low.

Within the three general intervention categories, policymakers can target specific drivers. For instance, when considering ChatGPT ethics, implementing restrictions on its usage can be a risk-reduction strategy by reducing exposure. Investing in awareness and responsible usage can increase resilience by driving transformative innovation, reducing vulnerability, and enhancing societal absorptive capacity. Generating new rewards can involve accessing new opportunities, such as developing AI detection tools or upskilling, as well as building new capabilities, like detecting and combating misinformation. The RRR framework offers nine potential interventions within these three categories, which can be pursued individually or in combination. Applying the RRR framework necessitates considering the distributional consequences of different interventions across actors, regions, and time. Risks and rewards are often unevenly distributed within a community, and pursuing immediate rewards may come at the cost of increased risks or decreased resilience in the future. Addressing distributional issues is crucial for ensuring fairness, sustainability, and the potential for meaningful change.

In the study, the RRR framework is employed to develop a simplified systems diagram that facilitates an understanding of the connections and policy choices relevant to addressing ChatGPT-related concerns. Figures 8–10 depict causal hypotheses, represented by arrows, indicating whether an increase in one factor leads to an increase or decrease in another. Green arrows denote increasing effects, while red arrows indicate decreasing effects. It is important to note that the existence and magnitude of these effects in practice are empirical questions that can be investigated. The examples and diagrams provided in the study, in accordance with the RRR framework (Roberts, 2023), serve as illustrations rather than comprehensive or empirical representations. They offer roadmaps for exploring empirical inquiries and considering normative trade-offs that policymakers will encounter and need to address.

Connections

When examining interconnection in relation to ChatGPT, it becomes evident that risk and reward often go hand in hand. For instance, providing students with access to ChatGPT presents an opportunity for increased efficiency in tasks such as text summarization and workload reduction (a). However, this also exposes them to risks such as plagiarism and cheating. Pursuing certain opportunities with ChatGPT, such as accessing vast amounts of information, can lead to rewards, but it also introduces risks like misinformation and copyright issues (b). Similarly, focusing on specific capabilities of ChatGPT, such as developing tools to detect plagiarism and misinformation, may enhance resilience in some areas (e.g., academic integrity). However, it may also create vulnerabilities in other domains, such as the digital divide, educational equity, and job losses (c). Institutions can play to their comparative advantage by recognizing the limitations of ChatGPT, addressing them through tools and strategies, and simultaneously transforming the existing teaching paradigm. Figure 8, with its green arrows (a, b, and c), illustrates these positive effects.

The mirroring of risk and reward sheds light on why policymakers often hold opposing views on the interdependence created by ChatGPT. Some may emphasize the rewards generated by the technology, while others may be more concerned about the risks associated with over-reliance on it or the neglect of curriculum development that promotes critical thinking and comprehensive learning. This dynamic is illustrated in Fig. 8. Regarding resilience, increasing knowledge and connectivity can enhance resilience in certain aspects while potentially creating challenges in others. Figure 9 demonstrates these multifaceted effects, showcasing the interplay between knowledge, connectivity, and resilience.

In advanced countries and institutions where knowledge acquisition, connectivity, and a willingness to embrace change are more prevalent, the nature of interventions required might be quite different (d). These entities have the capacity to absorb the ethical challenges posed by generative AI to a greater extent. On the other hand, countries and institutions with outdated teaching paradigms and resistance to technological shifts face greater difficulties in adapting to the era of generative AI (e). However, despite the negative impact of ChatGPT on educational practices and academic integrity, certain journals have responded effectively by implementing policies and developing technological tools to detect text generated by generative AI systems. Major publishers, such as Elsevier and Nature, have demonstrated a faster response compared to others, particularly in low-income countries. Their efficiency in addressing these challenges has given them a stronger financial position to handle the ethical conundrum posed by ChatGPT (f). Moreover, their access to global markets has provided them with more options to reach a wider readership (g).

Choices

Many higher education institutions are now actively assessing the vulnerability of their existing paradigms to disruption and considering critical models for their educational systems’ functioning. They are also exploring various strategic options to address the challenges posed by ChatGPT. Among the policy responses under consideration, legislation and restriction have been highlighted in the literature (Dwivedi et al., 2023; Lim et al., 2023). In this study, the RRR framework is utilized to map the intended aims of legislation policies and to identify their second-order effects, resulting in distinct “fingerprints” for the approach taken in each policy. Legislation plays a crucial role in organizing society and safeguarding the rights and responsibilities of individuals and authorities in relation to ChatGPT and similar technologies (De Jager, 2000). The legislation aims to enhance the absorptive capacity of institutions and stakeholders while also bolstering their adaptive capacity to effectively manage the impact of ChatGPT (Fig. 10).

The second-order effects of legislation regarding ChatGPT can have both positive and negative implications. One potential effect is a decrease in rewards due to the limitations imposed by the legislation, which may hinder individuals from fully capitalizing on the opportunities provided by ChatGPT (a). However, some individuals may be willing to bear this cost, particularly in terms of reducing workload, as it may contribute to a reduction in their vulnerability to negative effects on critical thinking and higher reasoning skills (b). While legislation can enhance resilience by increasing absorptive capacity, it may also have adverse effects on resilience in other aspects. For instance, if the legislation results in less efficient utilization of ChatGPT, it can lead to reduced rewards, which could undermine absorptive or adaptive capacity in dealing with other potential risks (c). In this scenario, a society may have well-designed legislation and responsible usage of ChatGPT, but it may show less interest in actively absorbing or adapting to additional responsibilities.

Summary

When assessing risk, reward, and resilience at an overall educational level, this study discussed the application of the RRR framework to the ethical concerns surrounding ChatGPT. The study explored the interconnected nature of risk, reward, and resilience and how they influence decision-making in various domains. The RRR framework provides a comprehensive and flexible mental model for policymakers to navigate complex problems. Moreover, the study examined the relationship between risk and reward in the context of ChatGPT, acknowledging that certain opportunities presented by the technology can also expose users to risks such as misinformation and copyright issues. Additionally, this study considered how focusing on specific capabilities of ChatGPT can introduce vulnerabilities in other areas, such as educational equity and the digital divide. Similarly, the importance of resilience in addressing the challenges posed by ChatGPT is discussed. While advanced institutions and countries with knowledge acquisition, connectivity, and adaptability have a higher capacity to absorb and respond to these challenges, others may struggle due to outdated teaching paradigms and limited technological resources and readiness.

Furthermore, the role of legislation in managing the impact of ChatGPT was examined within the RRR framework. This study recognized that legislation can have second-order effects, including a potential decrease in rewards by limiting the efficient use of ChatGPT. However, some individuals may be willing to accept these costs if it helps reduce their vulnerability to negative impacts on critical thinking skills. The study also discussed how legislation can enhance absorptive capacity but may simultaneously undermine adaptive capacity in the face of other risks. It was noted that a well-legislated society with responsible ChatGPT usage may exhibit less interest in absorbing or adapting to additional responsibilities. Hence, this study highlighted the importance of considering the interconnected nature of risk, reward, and resilience when addressing the ethical concerns associated with ChatGPT. The RRR framework provides policymakers with a valuable tool to navigate complex decision-making processes and consider the potential second-order effects of various policy interventions.

Discussion and implications

This study introduces a framework designed to assist policymakers and higher education institutions in addressing the complex challenges and opportunities presented by ChatGPT. The framework is specifically applied to tackle the ethical dilemmas associated with ChatGPT, with the aim of providing valuable insights to inform AI development policies. Firstly, the RRR is an integrative framework for policy or decision-making that is comprehensive and impartial, as noted by Roberts (2023). Specifically, there are other frameworks that focus on aspects like risk-and-reward (Terrile, Jackson & Belz, 2014; Ferràs-Hernández, 2023) and risk-and-resilience (Mochizuki et al., 2018). However, a notable limitation of RRR is its reluctance to provide guidance on how to assess the relative importance of different risks and rewards or how to balance them with considerations of resilience (Roberts, 2023). This is because such assessments necessitate standard judgments regarding what should be measured and also depend on empirical evidence regarding contextual facts and causal evaluations of the outcomes of various interventions. Instead of making standard judgments, RRR offers a simplified and structured systems model for navigating complex problems. It achieves this by identifying the drivers of each element, delineating their interconnections, elucidating the policy choices they enable, and specifying the consequences they yield, making assumptions that underpin the policy-making process. An alternative model capable of weighing different risks and rewards or balancing them against considerations of resilience can be derived from the field of multi-criteria decision-making (MCDM) literature, such as the analytical hierarchical process (AHP), as applied in previous research (Douligeris & Pereira, 1994; Malladi & Min, 2005; Zaidan et al., 2015, 2020; Sneesl et al., 2022a).

In particular, AHP is a widely employed technique within the MCDM literature, primarily used to address situations involving numerous criteria or factors, with the aim of resolving intricate challenges related to multi-criteria decision-making (Saaty, 1980, 1989; Douligeris & Pereira, 1994). The AHP methodology essentially breaks down a multi-criteria decision-making problem into a minimum of three hierarchical levels, encompassing the objectives, criteria, and decision alternatives, thus constructing a hierarchical model. It assesses the relative priorities of the criteria, conducts comparisons among the available decision alternatives for each criterion, and ultimately establishes a ranking of these alternatives. To determine the ranking of criteria or factors, the AHP relies on expert pair-wise comparisons, where judgments are expressed as “how much more one element dominates another concerning a specific attribute.” In contrast to AHP, RRR does not dictate what conclusions people and policymakers should draw when faced with complex problems. Instead, it offers a framework to guide individuals and groups in how to approach complex problems. RRR allows for the inclusion of complex and sometimes conflicting hypotheses on a single diagram, enabling experts from different disciplines to see that their insights and values are taken into account, while also making it evident that other experts bring different perspectives that require assessment and consideration. In this manner, RRR aids in the decision-making process, helping to identify the best course of action from the realm of feasible choices.

Secondly, several studies have called for more research on AI ethics and encouraged the development of policies to shape the development and utilization of AI tools in various sectors (Dwivedi et al., 2023; Mhlanga, 2023; Gunawan, 2023; Carvalho & Ivanov, 2023; Halaweh, 2023; Bukar et al., 2023; Tlili et al., 2023). In response to this demand, this study makes diverse and multidisciplinary contributions and holds implications and significance that are tailored to various audiences. Thus, Fig. 11 depicts the key stakeholders and the audience intended for this study. Firstly, policymakers responsible for regulating AI and emerging technologies will find this article valuable in understanding the ethical and legal implications of ChatGPT usage. This will help them formulate comprehensive policies and guidelines to ensure responsible AI integration and protect the interests of their constituents. Secondly, AI researchers and developers who work on ChatGPT and similar language models will benefit from the article’s insights into ethical considerations, data security, and accountability. It will guide them in creating AI systems that prioritize fairness, transparency, and privacy. Thirdly, business leaders and entrepreneurs looking to integrate ChatGPT into their products and services will gain valuable knowledge on navigating ethical challenges and ensuring user trust. The article will help them understand the importance of developing AI systems that align with ethical principles, ensuring long-term sustainability and customer satisfaction.

Figure 11 Beneficiary audience of the study.

Moreover, professionals working in the technology and AI industry will find the article relevant for staying informed about best practices and emerging trends related to AI policy and ethics. It will encourage them to advocate for responsible AI development within their organizations. In addition, scholars and educators focused on AI ethics and responsible AI will find the article a valuable resource to use in their research and teachings. It will serve as a reference for discussions on the social and ethical impact of AI and encourage further academic inquiry in this domain. Furthermore, while the article delves into technical and policy aspects, it is written in a way that makes it accessible to the general public and AI users. Individuals interested in AI’s impact on society, privacy, and ethics will gain a deeper understanding of ChatGPT’s implications and how it affects their daily lives. Hence, the article targets a broad audience encompassing decision-makers, professionals, researchers, and individuals with diverse backgrounds and interests in the ethical and policy aspects of ChatGPT usage. It aims to facilitate a meaningful and informed dialogue around responsible AI integration and inspire actions that prioritize ethical considerations in the development and deployment of AI systems.

Conclusion

The advent of Gen-AI marks a pivotal moment in human history, demanding a reevaluation of our coexistence with AI tools like ChatGPT. The integration of AI systems into our daily lives brings forth undeniable benefits but also introduces profound risks that could shape the future of civilization. This study has undertaken a comprehensive analysis of Gen-AI policy-making through the lens of the risk, reward, and resilience (RRR) framework, with a specific focus on ChatGPT. By meticulously identifying and categorizing key elements within the context of ChatGPT ethics, this study has showcased the intricate web of risks, rewards, and resilience factors that surround this technology. The RRR framework, a versatile and holistic approach, serves as a valuable tool for policymakers and higher education institutions grappling with the multifaceted dimensions of Gen-AI integration, particularly within the realm of higher education.

Key outcome

The application of the framework discusses the interconnected nature of risk and reward in the context of ChatGPT within higher education. The study highlighted that access to ChatGPT offers opportunities for efficiency, like text summarization and workload reduction, but also introduces risks such as plagiarism and cheating. Moreover, using ChatGPT to access vast information can yield rewards but bring risks like misinformation and copyright issues. Developing tools to detect plagiarism and misinformation can enhance academic integrity but may create vulnerabilities in areas like the digital divide and job losses. This risk-reward balance explains why policymakers hold opposing views on ChatGPT, some emphasizing its rewards and others worrying about the risks and neglect of critical thinking in education. Resilience can be increased in some aspects through knowledge and connectivity but can also pose challenges in others. In addition, the study also highlights the choices higher education institutions face in response to ChatGPT challenges, including considering legislation and restrictions. Legislation’s second-order effects can be both positive and negative, potentially decreasing rewards due to limitations but also reducing vulnerability to negative effects on critical thinking skills. Legislation can enhance absorptive capacity but may also affect resilience in various aspects, depending on its efficiency and societal attitudes towards responsibility.

Accordingly, through the application of Gen-AI to the context of higher education, this study demonstrates the RRR framework’s efficacy in addressing the concerns related to ChatGPT. It emphasizes the necessity of understanding the intertwined relationships between risk factors such as bias and misinformation, rewards including enhanced educational experiences, and resilience considerations like adaptability to emerging challenges. As AI technologies evolve, it becomes evident that effective policy responses must transcend simplistic solutions and instead embrace a holistic understanding of the intricate interactions between AI, society, and education. This study underscores the urgency of fostering collaboration among stakeholders—policymakers, educators, AI developers, and the public, to develop comprehensive and forward-looking policies that prioritize responsible and inclusive ChatGPT deployment. Therefore, in the pursuit of cultivating an environment where Gen-AI tools are harnessed to empower rather than undermine human progress, this research offers invaluable insights and guidance. As we navigate the ethical conundrums posed by ChatGPT, this study’s findings shed light on the ethical imperatives that guide our interactions with AI. Through a balanced consideration of risks, rewards, and resilience, we can pave the way for a future in which AI technologies enhance the fabric of higher education while preserving the foundations of a resilient and thriving society. Also, by embracing the lessons derived from this research, policymakers and higher education institutions can chart a course toward an AI-augmented future that is marked by responsibility, inclusivity, and collective wisdom.

Limitation and future research

This study is not without limitations, opening an avenue for future studies. Firstly, the primary focus of our article is the development and application of the RRR framework within the context of the ChatGPT ethical conundrum. The literature review served as a foundation to identify and extract the key elements necessary to practically demonstrate the dynamics of the RRR framework in a specific context, in this case, ChatGPT in higher education. This study assumed that the elements identified from the literature are relevant to the core elements of the framework (risk, reward, and resilience) to provide a clear, context-specific foundation for our research. However, the literature search was conducted in May 2023 and was limited to the Scopus database. This could present concern about the breadth of the literature review which is considered as a limitation of the current work. Therefore, future research could expand the literature search to encompass a broader range of sources. This can certainly explore the incorporation of additional, diverse sources in the literature review to offer a more comprehensive understanding of the topic and to further validate the application of the RRR framework. By doing so, future research could enhance the framework’s robustness and applicability across a wider range of scenarios.

Secondly, this study is subjective and conceptual and conducted in an effort to contribute to the theory regarding ChatGPT decision-making. Subjective research relies on the interpretation and analysis of observations, which can be difficult to quantify and measure objectively. Additionally, the researcher’s own biases can influence the results of the study. After populating the key elements into the RRR framework. The analysis of the ChatGPT ethics conundrum was done based on qualitative reasoning to demonstrate the applicability of the RRR framework to help stakeholders and decision-makers make an informed decision regarding chatGPT by evaluating its risk, reward, and resilience alongside each other. In the future, the RRR elements can be analyzed objectively through a robust and scientifically proven multi-criterion decision-making framework. In particular, future studies can utilize the AHP to formulate the problem into the decision matrix. This will help stakeholders and policy makers make transparent, efficient, and effective decisions and avoid biases frequently associated with qualitative reasoning. The output of the AHP can provide weight to the sub-elements under the three key classifications (risk, reward, and resilience). The decision can be reached by analyzing the weighting obtained by the analysis.

In addition, this study opened up a new direction for researchers to propose ethical frameworks and guiding rules for decision-making for RRR and other systems generally. The development of other ethically balanced rules as an extension can be explored in the future. The decision-making rules proposed in this study are only guidelines and there are bound to be exemptions in specific circumstances based on the certainty on whether rewards can truly be accrued, risks can be truly averted, or resilience can be achieved. It is essential to acknowledge these limitations in the present research. As a result, future research should aim to address this gap by devising strategies to mitigate the potential limitations. This could involve exploring alternative databases, methodologies, case studies, or theories.

Appendix

Critical analysis of risk elements

Refer to Table A1.

Critical analysis of reward elements

Refer to Table A2.

Critical analysis of resilience elements

Refer to Table A3.

Supplemental Information

Supplemental Information 1 Data and Critical Analysis of the Extracted Information.

The extracted data obtained from the articles reviewed as well as the critical analysis of the data based on themes of the policy making framework proposed in this study,

Additional Information and Declarations

Competing Interests

Author Contributions

Data Availability

The authors declare that they have no competing interests.

Umar Ali Bukar conceived and designed the experiments, analyzed the data, prepared figures and/or tables, authored or reviewed drafts of the article, and approved the final draft.

Md Shohel Sayeed conceived and designed the experiments, prepared figures and/or tables, authored or reviewed drafts of the article, and approved the final draft.

Siti Fatimah Abdul Razak performed the experiments, prepared figures and/or tables, authored or reviewed drafts of the article, and approved the final draft.

Sumendra Yogarayan performed the experiments, analyzed the data, prepared figures and/or tables, authored or reviewed drafts of the article, and approved the final draft.

Oluwatosin Ahmed Amodu analyzed the data, authored or reviewed drafts of the article, and approved the final draft.

The following information was supplied regarding data availability:

This is a literature review.

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
