# Peer review of "An integrative decision-making framework to guide policies on regulating ChatGPT usage"

_PeerJ Computer Science, doi:10.7717/peerj-cs.1845_

## Round 0.1 · original submission · Major Revisions

The description of the methodology should definitely be more precise: reproducibility of a SLR asks for more. Also, concerns about limitations of the approach should be specifically addressed, among the other concerns expressed by reviewers.

**Language Note:** PeerJ staff have identified that the English language needs to be improved. When you prepare your next revision, please either (i) have a colleague who is proficient in English and familiar with the subject matter review your manuscript, or (ii) contact a professional editing service to review your manuscript. PeerJ can provide language editing services - you can contact us at copyediting@peerj.com for pricing (be sure to provide your manuscript number and title). – PeerJ Staff

Reviewer 1 ·

Basic reporting

1. The author's decision to incorporate 41 articles in the abstract is commendable, but it would greatly enhance the clarity of their intent if they provided a clear rationale for this specific number. Are there specific criteria or significance thresholds associated with these articles that influenced their selection?
2. While the abstract provides a useful overview, it would be beneficial to incorporate some of the key findings into this section. This would give readers a glimpse of the study's outcomes and entice them to explore further.
3. It's important to address the use of blue-colored paragraphs in the text. Explaining the significance or purpose behind this formatting choice would help readers understand if it serves as a visual aid, highlights key points, or serves another purpose.
4. To enhance the engagement of readers, it would be beneficial to include visuals in the introduction section. Visual elements can serve as an effective way to convey complex information or draw attention to critical concepts.
5. While the authors propose an integrative framework, it's essential to acknowledge the possibility of alternative solutions. Discussing potential alternatives and their pros and cons would demonstrate a comprehensive understanding of the problem space.
6. Highlighting the unique contributions of the authors in applying the RRR framework within the context of ChatGPT is crucial. Readers should be explicitly informed about how this framework adds value to the study.
7. To bolster the literature review, the authors should consider incorporating additional relevant papers. Expanding the review to encompass a broader range of sources would provide a more comprehensive understanding of the topic.

Experimental design

8. The methodology section could benefit from additional details to ensure transparency and reproducibility. Clarifying the research methods, data sources, and any potential limitations would strengthen the study's rigor.
9. Including specific evaluation metrics in the study is imperative. These metrics would allow readers to assess the validity of the authors' conclusions and provide a means for replication or further research.

Validity of the findings

10. The authors should strive for clarity and coherence in their writing. Explicitly articulating their research objectives, methodologies, and validation processes will enhance the overall quality of the paper, making it more accessible and informative to the audience.
11. It would be advisable for the authors to incorporate a section addressing the constraints and shortcomings of their research.

Reviewer 2 ·

Basic reporting

This reviewer’s suggestions, questions, and concerns for the paper are listed below:

1. The motivation of the paper does not exist. The abstract does not clearly explain the contribution. The importance and contribution are not highlighted
2. Complete form of abbreviations may be used the in the abstract.
3. Keywords should be listed in alphabetical order.
4. The limitation(s) of methodology used in this review should be discussed. Other possible methodologies that can be used to achieve the objective in relation in this work should also be analyzed.
5. The authors are supposed to focus on the main topic of the study and present a Literature Review in the form of tables in order to make research gaps and innovations easy to detect. Authoritative synthesis assessing the current state-of-the-art is absent. In the literature review section, the focus should be on a critical analysis of the gradual advancement, as well as the current level, of the state-of-the-art, with quantitative information on the time & space complexity, as well as on the accuracy obtained by each cited methodology. The advancement offered by each cited methodology should be made clear. Authors cite many related works but then none of them is included in the experimental comparison.
6. The limitation(s) of methodology used in this review should be discussed. Other possible methodologies that can be used to achieve the objective in relation in this work should also be analyzed.
7. The Conclusion section of the paper is indicative, but it could be improved to emphasize the significance and relevance of the work done with some more detailed considerations, to wrap up the obtained results and provide a reference point for readers.

Experimental design

Please see basic reporting.

Validity of the findings

Please see basic reporting.

Additional comments

Please see basic reporting.

·

Basic reporting

This study is needed for the current AI world.

Experimental design

The study design is good.

Validity of the findings

The findings of the study may be useful to the policymakers.

Additional comments

Abstract:
• Please mention the aim and adopted methodology of the study.
Introduction:
• Provide the novelty of the study in the Introduction.
• The authors may give the expanded form of RRR while using at first in line number – 87
OVERVIEW OF RRR:
• This section is much limited. The authors may expand it for detailed insights.
Methodology:
• Why did the authors focus only on Scopus?
Sections 4 and 5 have been drafted well.

6.2 Future research, this section may be moved below to the conclusion section.

What are the limitations of the study?

The following reference can be cited in this paper as it seems relevant to the present research.
Ágnes Kemendi , Pál Michelberger , Agata Mesjasz-Lech. INDUSTRY 4.0 AND 5.0 – ORGANIZATIONAL AND COMPETENCY CHALLENGES OF ENTERPRISES. Polish Journal of Management Studies 2022; 26 (2): 209-232. DOI: 10.17512/pjms.2022.26.2.13

---

## Round 0.2 · accepted · Accept

The authors addressed the reviewers' concerns, and the paper is ready for publication.

Reviewer 1 ·

Basic reporting

All changes have been done

Experimental design

All changes have been done

Validity of the findings

All changes have been done

Reviewer 2 ·

Basic reporting

The reviewer has carefully evaluated the authors' responses and their actions in the revised version of the paper. This reviewers found them persuasive and applicable. Hence, the revised paper can be accepted for publication.

Experimental design

The paper is updated upon the reviewer comments.

Validity of the findings

The paper is updated upon the reviewer comments.

Additional comments

Authors responses covered the reviewers concerns.

·

Basic reporting

No Comment

Experimental design

No Comment

Validity of the findings

No Comment

Additional comments

The authors have revised the paper as needed and requested. The current version of the paper can be considered for publication based on the journal's policy.